# New Insights into Immunological Involvement in Congenital Disorders of Glycosylation (CDG) from a People-Centric Approach

**DOI:** 10.3390/jcm9072092

**Published:** 2020-07-03

**Authors:** Rita Francisco, Carlota Pascoal, Dorinda Marques-da-Silva, Sandra Brasil, Fernando M. Pimentel-Santos, Ruqaiah Altassan, Jaak Jaeken, Ana Rita Grosso, Vanessa dos Reis Ferreira, Paula A. Videira

**Affiliations:** 1CDG & Allies—Professionals and Patient Associations International Network (CDG & Allies-PPAIN), Departamento Ciências da Vida, Faculdade de Ciências e Tecnologia, Universidade NOVA de Lisboa, Caparica, 2825-149 Lisbon, Portugal; rab.francisco@campus.fct.unl.pt (R.F.); cm.pascoal@campus.fct.unl.pt (C.P.); dorinda.silva@ipleiria.pt (D.M.-d.-S.); sd.brasil@fct.unl.pt (S.B.); pimentel.santos@nms.unl.pt (F.M.P.-S.); jaak.jaeken@kuleuven.be (J.J.); sindromecdg@gmail.com (V.d.R.F.); 2UCIBIO, Departamento Ciências da Vida, Faculdade de Ciências e Tecnologia, Universidade NOVA de Lisboa Caparica, Caparica, 2825-149 Lisbon, Portugal; ar.grosso@fct.unl.pt; 3Portuguese Association for Congenital Disorders of Glycosylation (CDG), Departamento Ciências da Vida, Faculdade de Ciências e Tecnologia, Universidade NOVA de Lisboa, Caparica, 2825-149 Lisbon, Portugal; 4School of Technology and Management, Polytechnic Institute of Leiria, 2411-901 Leiria, Portugal; 5CEDOC, Faculdade de Medicina, Universidade NOVA de Lisboa, Campus Hospital Egas Moniz, 1349-019 Lisbon, Portugal; 6Department of Medical Genetic, King Faisal Specialist Hospital and Research Center, Riyadh 12713, Saudi Arabia; ruqaiah.altassan@mail.mcgill.ca; 7Center for Metabolic Diseases, Department of Pediatrics, KU Leuven, 3000 Leuven, Belgium

**Keywords:** congenital disorder(s) of glycosylation (CDG), PMM2-CDG, immune response, infections, allergies, vaccination, gastrointestinal tract (GI), e-questionnaire, people-centricity

## Abstract

Congenital disorders of glycosylation (CDG) are rare diseases with variable phenotypes and severity. Immunological involvement remains a largely uncharted topic in CDG, mainly due to lack of robust data. To better characterize immune-related manifestations’ prevalence, relevance, and quality-of-life (QoL) impact, we developed electronic questionnaires targeting (1) CDG patients and (2) the general “healthy” population. Two-hundred and nine CDG patients/caregivers and 349 healthy participants were included in this study. PMM2-CDG was the most represented CDG (*n* = 122/209). About half of these participants (*n* = 65/122) described relevant infections with a noteworthy prevalence of those affecting the gastrointestinal tract (GI) (63.1%, *n* = 41/65). Infection burden and QoL impact were shown as infections correlated with more severe clinical phenotypes and with a set of relevant non-immune PMM2-CDG signs. Autoimmune diseases had only a marginal presence in PMM2-CDG (2.5%, *n* = 3/122), all being GI-related. Allergy prevalence was also low in PMM2-CDG (33%, *n* = 41/122) except for food allergies (26.8%, *n* = 11/41, of PMM2-CDG and 10.8%, *n* = 17/158, of controls). High vaccination compliance with greater perceived ineffectiveness (28.3%, *n* = 17/60) and more severe adverse reactions were described in PMM2-CDG. This people-centric approach not only confirmed literature findings, but created new insights into immunological involvement in CDG, namely by highlighting the possible link between the immune and GI systems in PMM2-CDG. Finally, our results emphasized the importance of patient/caregiver knowledge and raised several red flags about immunological management.

## 1. Introduction

Congenital disorders of glycosylation (CDG) are a diverse and rapidly expanding family of more than 140 genetic diseases caused by defects in the cellular machinery responsible for assembling, trimming, and adding glycans to proteins and lipids. Most CDG are defects in N- glycosylation, O-glycosylation or both [1,2,3]. Defects in the glycosylphosphatidylinositol (GPI)-anchor synthesis have more recently been added to this family [4]. CDG are mainly complex, multi-system disorders which—despite recent exciting therapeutic developments—still widely lack effective treatments [5,6,7,8,9,10]. PMM2-CDG (MIM: 212065) was the first reported N-glycosylation defect and it is the most common type with approximately 1000 patients reported worldwide [5,11,12]. It shows a broad spectrum of clinical signs and phenotypic severity both among different patients and within the same patient throughout life. Neurological involvement—particularly hypotonia, ataxia and intellectual disability—is a hallmark of this (and many other) CDG. Other systems/organs are usually also affected, including the gastrointestinal tract (GI) and the immune system [13,14,15].

Some CDG have a fairly well-defined immunological phenotype (e.g., PGM3-CDG (MIM: 615816), MOGS-CDG (MIM: 606056), SLC35C1-CDG (MIM: 266265)) [14,16]. A new addition to this list is MAN2B2-CDG which has an immunodeficiency phenotype [1]. Regarding PMM2-CDG, infections are a major contributor to the 20% infantile mortality [17]. Moreover, in some patients, infections have been identified as triggers of other clinical signs, such as stroke-like episodes (SLE) [18]. Besides relevant infections, there are reports of allergic manifestations and to a lesser extent of autoimmune diseases in CDG [14,16]. Altered and/or inefficient vaccination responses have also been described [13,17]. From a biological standpoint, these clinical descriptions can be explained by the glycans’ and glycosylation’s central role in immune pathways, particularly in cell–cell interaction and pathogen recognition, as well as in immune response modulation [14,19]. However, as most CDG are associated with numerous anatomical and functional defects in various organs, including the immune system, the immune pathophysiology of these patients may be highly complex and multi-factorial.

Patient or people centricity could be explained as a dynamic co-creation process which involves the patients/citizens as full partners by continuously promoting their empowerment and engagement and by valuing their insights, preferences, values, and beliefs [20]. Families/patients, particularly in rare diseases, are uniquely positioned to provide data on health-related topics, as they experience and permanently deal with the disease, becoming experts in their own condition [21]. Indeed, people-centricity has redefined needs and priorities in CDG and may still open new and (more) meaningful research avenues [22].

The widespread use of the Internet and social media has created promising opportunities for patient-centric research [23,24]. Among them are the use of electronic questionnaires (e-questionnaires). These tools are particularly relevant in rare diseases such as CDG where patients are geographically dispersed, and investment is scarce.

### Study Rationale and Aims

Due to glycan relevance in the immune response and clinical evidence suggestive of immunological dysfunction in CDG, but lack of a clear immune-clinical portrait of CDG patients, we decided to build a comprehensive e-questionnaire targeting CDG patients and caregivers—the Immunology and CDG questionnaire (ImmunoCDGQ). This tool is based on literature revision and input from CDG clinical experts and families. It addresses a set of immune-related topics, including infections, allergies, autoimmunity, and vaccination response.

Primary aim: To better understand immunological (dys)function in CDG, focusing on PMM2-CDG, the most prevalent type. With this purpose we have:(1)characterized the infection profile of PMM2-CDG patients, including infection types and frequency, age association, and treatment effectiveness;(2)studied allergies (types, prevalence, association with infection and age);(3)evaluated autoimmune diseases (types, prevalence, infection and age relation);(4)examined vaccinations (adherence, perceived (in)effectiveness and relevant adverse reactions (ADRs);(5)assessed a potential correlation between genotype and (immune) phenotype, as well as correlated overall phenotypic severity and other clinical manifestations with immune-related manifestations;(6)measured the impact of immune-related manifestation on the quality of life (QoL) of these patients.

Secondary aim: To untap information only patients/caregivers hold, we have employed a people-centric research approach. Hence, to assess whether “non-experts” (e.g., patients) are reliable health-data providers, and to evaluate the relevance of people-driven research is an important additional aim of this study.

## 2. Experimental Section

### 2.1. Development of the ImmunoCDGQ and ImmunoHealthyQ (Immunology Questionnaire for the General “Healthy” Population)

The scientific/clinical content and structural organization of the ImmunoCDGQ was based on: (1) a literature review on immunological involvement in CDG by our group [16]; (2) the input from the medical/scientific and CDG family advisory committees, and (3) questionnaire-based tools to assess immune-related clinical manifestations [25,26,27,28,29]. These tools were characterized and selected based on the following criteria: generic/disease-specific questionnaire, main domains, specific domains, mode of administration, number of items, score system, completion time, available translations (cross-cultural studies) and validated tools. The ImmunoCDGQ comprises the following sections: Patient Information (demographics), CDG immunology checklist, CDG immune problems, Infection history, Autoimmune history, Allergy history, Vaccination history, Quality of life, and CDG symptom checklist. Three questions were added to the end of the questionnaire to evaluate its understandability, the use and usefulness of the glossaries provided, and to assess the experience of participants. The ImmunoHealthyQ was a version of the ImmunoCDGQ for the general “healthy” population.

### 2.2. Piloting, Translations, and Refinement by Clinical and Family Experts

The ImmunoCDGQ was originally developed in English and tested in two pilot phases by CDG families and expert clinicians/researchers. In total, 11 experts participated in both pilot rounds. The questionnaires were translated into five additional languages: French, Spanish, Portuguese, Italian and Arabic. All translations were performed and/or reviewed by clinical experts who are native speakers of each language.

### 2.3. Ethical Approval

The project received a favorable opinion from the ethical committee of Egas Moniz Hospital (n° 20170700050 on 23/04/2018). E-informed consent was obtained from all the participants.

### 2.4. Survey Monkey—E-Questionnaire Platform

An e-format—using the Survey Monkey platform (http://www.surveymonkey.net—Copyright #1999–2019 SurveyMonkey)—was adopted for the questionnaires taking advantage of a methodology already implemented by our group [21]. The respondents’ IP identifier was not recorded to assure participant anonymity. Restriction of multiple entries from the same device was set to avoid duplication. Different question formats such as multiple choice, matrix, and open-ended questions were used. Importantly, Survey Monkey also allowed the addition of logic to questions. Thus, despite the total number of items in the ImmunoCDGQ being 58 (56 items in the ImmunoHealthyQ), participants were directed to specific questions/sections according to previous answers. This decreased participant burden and guided participation. On average (in the English version), the ImmunoCDGQ was completed in 30 min (the ImmunoHealthyQ in 10 min). An electronic copy of the ImmunoCDGQ (in all language versions) is available at https://www.researchcdg.com/immunocdgq.html, while the ImmunoHealthyQ (also all languages) is publicly available at https://www.researchcdg.com/immunoq.html.

### 2.5. Information and Recruitment Strategy

The questionnaires were launched online on 1 October 2018 and stayed open for participation till 24 June 2019. A multi-platform and multi-stakeholder approach was adopted to overcome geographic limitations, maximize participation, and prevent sampling bias. Prior to the questionnaires’ launch, materials dedicated to immunology were made freely available and disseminated by direct emailing and social media. All social media channels used in the ImmunoCDGQ recruitment campaign were managed by the Portuguese Association for CDG (APCDG), which has a consolidated stand in the CDG community. Other questionnaires, awareness and educational materials promoted through these media have generated great responsiveness and engagement [21]. The European reference network for rare metabolic diseases (MetabERN) also endorsed this study and assisted in its communication. Recruitment for the ImmunoHealthyQ mandatorily differed from the strategy developed for the ImmunoCDGQ due to differences in the target audience. Appendix A lists all the dissemination channels used in the recruitment strategies.

### 2.6. Inclusion/Exclusion Criteria

Given the scientific/medical and structural construction of the ImmunoCDGQ, and thanks to the targeted dissemination strategy developed, “random” participation of non-CDG patients/caregivers was considered highly unlikely.

All CDG patients reported in the ImmunoCDGQ with confirmed CDG diagnosis were included. Patients without a confirmed CDG diagnosis were excluded. Possible patient duplications were carefully screened and when in doubt both reports were excluded. For the control group, only participants not reporting chronic/genetic conditions were included. Since only healthy participants could be included in the control all ImmunoHealthyQ participants identifying autoimmune conditions were excluded, serving the Autoimmune history section of the ImmunoHealthyQ only for inclusion/exclusion purposes. All patients and participants failing to complete all relevant clinical sections were eliminated from further analysis.

### 2.7. Statistical Analysis

Statistical analysis was done using the R programming language on RStudio software version 3.6.1 and graphs were created on GraphPad Prism version 7.0. Descriptive statistics were performed to analyze and report findings. Statistical significance was calculated using Fisher’s exact test, since all variables were categorical with the majority being binomial. Power test analysis—calculated with the pwr.2p2n.test function from pwr R package [30]—revealed that a sample size of 55 PMM2-CDG patients achieved a power of 95% and a level of significance of 5% (two-sided) for detecting a medium effect size of 0.5 between paired observations. We considered the worldwide PMM2-CDG patient population to be of 1000 individuals [7,12]. The calculated sampling power for comparisons between the PMM2-CDG (*n* = 122) and control cohorts (*n* = 349) was of 99% with a level of significance of 5% (two-sided) for detecting a medium effect size of 0.5 between paired observations. Statistical significance was defined as *p* value ≤ 0.05 (*). Odds ratio (OR) values are also presented.

### 2.8. Genotype Characterization

In the Patient information (demographics) section there was the possibility for participants to share CDG variants. The group of PMM2-CDG patients who shared variants (*n* = 66/122) was thoroughly explored through genotype characterization. Such analysis aimed to: (1) confirm the putative clinical relevance (pathogenicity) of each mutation; and (2) inquire whether any genotype-phenotype associations (e.g., immune manifestations or overall phenotypic severity) could be established. Variants reported by the participants were searched for in the following human genomic databases:Human Genome Mutation Database (HGMD ^®^, http://www.hgmd.cf.ac.uk/) [31];Leiden Open Variation Database 3.0 (LOVD ^3^, https://databases.lovd.nl) [32];Expasy (https://www.expasy.org/) [33];ClinVar (NCBI, https://www.ncbi.nlm.nih.gov/clinvar/) [34];Varsome clinical (https://varsome.com/) [35].

An annotated classification of each variant was extracted from these databases (Appendix A). Variants were considered as “novel” when they were absent from all the analyzed databases and the literature. Variant pathogenicity was assessed though computational frameworks. The threshold values used to classify variants as damaging were defined based on previous literature (Appendix A). Briefly, the *Homo sapiens* reference genome (hg19) NCBI 37 was selected as the default reference for uniformity and consistency reasons, since hg19 was transversally present in all the prediction tools used. To get the genomic location of previously unreported variants, the position converter tool from the Mutalyzer platform (https://mutalyzer.nl/) was selected. For all annotated variants, the genomic location provided on dbSNP NCBI (https://www.ncbi.nlm.nih.gov/snp/?cmd=search) was considered. The platform Genomizer (http://genomizer.com/) which functions as a hub of in silico pathogenicity prediction tools and databases was used to optimize and accelerate analysis with multiple tools.

## 3. Results

### 3.1. Participants Display Diversified Age, Gender and Worldwide Distribution

Our questionnaires assessed 209 CDG patients and 349 healthy participants. PMM2-CDG was the most represented CDG (58.4%, *n* = 122/209) and included 4 deceased patients (Appendix A). In total 35 different CDG were included. After PMM2-CDG, ALG6-CDG (MIM:603147) (4.3%, *n* = 9/209) and SLC35A2-CDG (MIM: 300896) (3.3%, *n* = 7/209) were the most reported (Figure 1A). This prevalence echoes published data on these CDG [6,12,36]. All these CDG were clustered and herein referred to as the non-PMM2-CDG group (Figure 1A). The study had a diverse distribution of participants amid the control, PMM2-CDG and non-PMM2-CDG groups concerning age (Figure 1B), gender (Figure 1C) and geographic distribution (Figure 1D). Most CDG patients were diagnosed before 5 years of age (74.2%, *n* = 155/209) (Appendix A). Only 17.2% of PMM2-CDG patients (*n* = 21/122) received a diagnosis from 6-years-old onwards, whereas among non-PMM2-CDG patients this percentage was 37.9% (*n* = 33/87). Of note, 4.1% of the PMM2-CDG patients (*n* = 5/122) were diagnosed from 25-years-old onwards. No diagnoses were made after 25 years of age in non-PMM2-CDG (Appendix A). This reflects the growing adult PMM2-CDG population, patient phenotypic heterogeneity and probably the differences in access to diagnosis across countries [37,38]. Mothers were the most common ImmunoCDGQ respondents (in 82.8%, PMM2-CDG—*n* = 101/122—and in 87.4% non-PMM2-CDG—*n* = 76/87). This is similar to what is described in other rare disease studies [21], [39]. In the ImmunoHealthyQ, most participants reported themselves (55.9%, *n* = 195/349) (Appendix A).

### 3.2. CDG Patients Show a Higher Prevalence of Immune-Related Manifestations

To better understand CDG patients’ immune (dys)function, we compared immune-related manifestations and their prevalence in CDG relative to the control group. According to their immune-related reports, participants were assigned to one of three categories: (1) with immunological involvement, (2) without immunological involvement, or (3) unsolved immune status. When we consider the prevalence of immunological involvement in each of the study groups, the prevalence was of 67.2% in PMM2-CDG (*n* = 82/122), 80.5% in non-PMM2-CDG (*n* = 70/87), and 52.2% in the control (*n* = 182/349) (Figure 2A). Noteworthy, in all the study groups, a residual percentage of participants were classified as having an unsolved immune status, i.e., uncertainty about the participant’s immunological status or no clear immune-related manifestations reported. Examples are participants with unexplained frequent fevers, laboratory alterations and/or vaccination inefficacy in the absence of other immune manifestations. Compared to the control (2.3%, *n* = 8/349) and non-PMM2-CDG groups (2.3%, *n* = 2/87), there were more patients with an unsolved immune status in the PMM2-CDG group (10.7%, *n* = 13/122) (Figure 2A). All these participants were withheld from further analysis.

Regarding the types of immune clinical features identified by the participants with immunological involvement, relevant infections were the most described in PMM2-CDG and non-PMM2-CDG patients, while allergies accounted for most immune-related reports among controls (Figure 2B).

Literature reports of autoimmune manifestations, particularly affecting the GI tract, have been made in CDG [14,16]. In our CDG group, autoimmune diseases had a marginal prevalence (Figure 2B). Only 3.8% of all CDG patients (*n* = 8/209)—3 PMM2-CDG and 5 non-PMM2-CDG—reported a confirmed diagnosis (Appendix A). Most of these patients (62.5%, *n* = 5/8) reported GI-related autoimmune manifestations, such as celiac disease (*n* = 3/8), inflammatory bowel disease (*n* = 1/8) and eosinophilic esophagitis (*n* = 1/8). An additional PMM2-CDG participant reported that celiac disease was under study.

Dietary supplementation (e.g., specific sugars and trace elements) and their effect on immune (dys)function was assessed in our study due to the increased interest in these approaches for different CDG, including its use as an immunotherapy [9,40,41]. The very few PMM2-CDG participants with immunological involvement on dietary supplementation therapies reported little to no positive impact on immune manifestations. Comparatively, more non-PMM2-CDG participants used these therapeutic strategies and evaluated them as more effective at improving immune dysfunctions (Appendix A).

Because of the existing intra- and interclinical heterogeneity among the non-PMM2-CDG group (Appendix A) and since PMM2-CDG was the most represented, the following sections of the paper are mainly dedicated to PMM2-CDG. Although no statistical comparisons of clinical features were performed for the non-PMM2-CDG group, it was referred to, when appropriate, to help frame the PMM2-CDG group.

### 3.3. PMM2-CDG Patients Present Diverse and Severe Infection Patterns Which Are an Important Hospitalization Cause

Few and often undetailed descriptions of recurrent and/or severe infections in CDG (including PMM2-CDG) exist [13,14,16,17]. To better clarify and characterize CDG patients’ infections, a questionnaire section was dedicated to exploring their prevalence, most affected organs, and their frequency/severity.

Prevalence of relevant infections among PMM2-CDG patients was of 53.3% (*n* = 65/122), while only 15.2% (*n* = 53/349) of the control participants described any relevant infection-related issues (Appendix A). When questioned about the frequency and severity of infections, PMM2-CDG participants frequently described infectious episodes as severe and/or triggers of other clinical manifestations (84.6%, *n* = 55/65). Additionally, PMM2-CDG infections exhibited higher complexity and a more considerable overlap between high frequency and great severity when compared to the control group who mostly labelled infections solely as recurrent (75.5%, *n* = 40/53) (Appendix A). As for the non-PMM2-CDG group, although participants described infections mainly as recurrent (77.2%, *n* = 44/57), an intricate pattern of frequency/severity resembling that of PMM2-CDG was also observed (Appendix A).

To better characterize infections and their associated issues, questions about the present/recent past were included. At the time of participation in the study and excluding three PMM2-CDG patients who had died, similar percentages of PMM2-CDG (72.6%, *n* = 45/62) and control participants (71.7%, *n* = 38/53) reported infections. However, 82.2% of these PMM2-CDG patients (*n* = 37/45) were reported to have multiple—affecting several organs—infections. This was significantly higher than in the control group (52.6%, *n* = 20/38) (Figure 3A). A comparison by organ-specific infections between PMM2-CDG and controls only showed a significant statistical difference for GI infections (24.2%, *n* = 15/62 of PMM2-CDG and 9.4%, *n* = 5/53 of control participants, *p* = 0.48 and OR = 3.1). However, as shown in Appendix A, lung infections (12.9%, *n* = 8/62), otitis (22.6%, *n* = 14/62) and cold/flu (51.6%, *n* = 32/62) were all also more prevalent in PMM2-CDG. Additionally, heart (3.2%, *n* = 2/62), liver infections and sepsis (1.6%, *n* = 1/62 each) were only present in PMM2-CDG. As for respiratory tract (RT) infections, these were slightly lower in PMM2-CDG patients (33.9%, *n* = 21/62) when compared to the control participants (38.8%, *n* = 19/53) (Appendix A).

Considering infection prevalence and related hospitalizations in the past 12 months, PMM2-CDG patients showed a significantly higher number of infections (62.5%, *n* = 30/48, reporting >3 infections) and of infection-driven hospitalizations (41.9%, *n* = 26/62) (Figure 3B,C). Accordingly, most PMM2-CDG participants identified infections as the main hospitalization cause (71.7%, *n* = 43/60) (Figure 3D). In 44.2% of these PMM2-CDG patients (*n* = 19/43) infections were recognized to be the main cause of hospitalizations, while the remaining 55.8% (*n* = 24/43) considered infection-caused hospitalizations a major cause of concern when the patients were younger. In the control group, 47.2% of the participants (*n* = 25/28) also identified infections as the main hospitalization cause, but this was only true mainly when the participants were younger (56%, *n* = 14/25).

### 3.4. Infections in PMM2-CDG Have Greater Clinical Relevance in Infancy and Significantly Associate with the GI Tract

To further define how age modulated infections, participants were asked to rate the impact of ageing on infections. Most PMM2-CDG (70.8%, *n* = 46/65) and control participants (69.8%, *n* = 37/53) showed a tendency to consider that age improved infection recurrence and severity. This trend was slightly higher in PMM2-CDG both in terms of effect size and magnitude, but no significant differences were found (Appendix A). Particularly in PMM2-CDG, birth to 3 years was significantly more recognized as the time when infections had had greater recurrence and/or severity (40%, *n* = 24/60) (Figure 3E). This reveals that relevant infections generally do not extend into childhood as previously suggested [13,42]. Despite these results, no meaningful differences were reported in this questionnaire about the perception of acquired/increased immune resistance between the PMM2-CDG and control groups.

Participants were then asked to identify the types (by affected organ) and frequency (participants could choose from 4 options: single occurrences, sporadic, frequent or chronic) of the infections they had experienced during the time selected as the period of their lives when infections had the greatest clinical significance. The identified infections differed between PMM2-CDG and control groups (Appendix A), though mainly consistent with the types of infections reported for both groups at present/in the recent past (Appendix A).

Regarding overall infection frequency, fewer PMM2-CDG patients (70.8%, *n* = 46/65) were reported to have suffered from at least one frequent/chronic infection when compared to control participants (84.9%, *n* = 45/53). Despite this, more PMM2-CDG participants (69.6%, *n* = 32/46) described multiple (≥2) frequent/chronic infections than controls (62.2%, *n* = 28/45) (Appendix A).

Relating to infection prevalence by affected organ/system, GI and tooth/mouth infections emerged in terms of total prevalence and/or recurrence. GI infections were found to have a higher total prevalence in PMM2-CDG (63.1%, *n* = 41/65) than in the control group (50.9%, *n* = 27/53), frequent/chronic GI infections being significantly higher in PMM2-CDG (42.5%, *n* = 17/40) (Figure 3F). Oppositely, tooth cavities/abscesses and mouth infections were less common among PMM2-CDG (26.2%, *n* = 17/65) than in control participants (54.7%, *n* = 29/53). In fact, both frequent/chronic (7.8%, *n* = 4/51) and sporadic (21.7%, *n* = 13/60) prevalence were statistically lower in PMM2-CDG (Figure 3G and Appendix A).

As for respiratory (including cold/flu, otitis, and lung infections) infections, they were, when considered globally, the most reported in PMM2-CDG and control participants. Regarding their recurrence, frequent/chronic RT infections were less common in PMM2-CDG (33.8%, *n* = 22/65) than in control participants (54.7%, *n* = 29/53), but this difference did not reach statistical significance (Appendix A).

Rarely infection affected organs included the heart, liver, and brain. Blood infections also had a low prevalence with frequent/chronic infections being exclusive of PMM2-CDG (3.6%, *n* = 2/55). One-time/sporadic liver infections (17.3%, *n* = 9/52) and blood infections (18.9%, *n* = 10/53) were considerably more frequent in PMM2-CDG (Appendix A).

Frequent, severe, and chronic infections can lead to permanent damage. Despite the severe, intricate, and diverse infection patterns observed in PMM2-CDG, permanent infection damage was uncommon (38.5%, *n* = 25/65) (Appendix A).

### 3.5. PMM2-CDG Infections Do Not Relate to a Specific Recovery Time, Season, or Infectious Agent, but Significantly Correlate with Frequent Fever Episodes and Overall Phenotypic Severity

We also aimed to characterize PMM2-CDG infections as to their recovery time, possible season association, and preponderant causal agent(s). Time to cure an infection did not differ substantially between PMM2-CDG and controls (62.3%, *n* = 33/53 took ≥1–2 weeks), although PMM2-CDG patients (73.8%, *n* = 48/65 took ≥1–2 weeks) showed a tendency to take a longer time to get rid of infections (OR = 1.93) (Appendix A). Moreover, no clear season-dependent or independent relationship between infections was found in our dataset. Still, 63.1% of the PMM2-CDG (*n* = 41/65) participants asserted only partial to no season-infection association (Appendix A). Relating to infectious agent(s), no strong association with or preponderance of an agent(s) was found in our dataset, especially in PMM2-CDG. However, PMM2-CDG participants (27.7%, *n* = 18/65) identified virus infections more frequently than controls (18.9%, *n* = 10/53) (Appendix A).

Fever is a common sign of infection. Frequent unexplained fever episodes (FUFEs) have been described in PMM2-CDG [13,43]. In our study FUFEs were found to be significantly more common in PMM2-CDG (42.7%, *n* = 35/122) than in the control group (6.6%, *n* = 23/349) (*p* = 1.15 × 10^−9^/OR = 5.95). Hyperthermia of unknown origin has been observed in PMM2-CDG [13,18], explaining at least partly these differences. However, FUFEs in PMM2-CDG (42.9%, *n* = 27/63) were strongly related to infections (*p* value = 1.55 × 10^−5^/odds ratio = 7.62) (Figure 3H). Comparison of FUFE prevalence among PMM2-CDG (7.4%, *n* = 2/27) and control participants (4.4%, *n* = 7/158) without immunological involvement (i.e., relevant infections, allergies or autoimmune diseases) did not retrieve any significant differences, further strengthening the link between FUFEs and an immunological phenotype. The majority of CDG have a broad range of clinical phenotypes and severity [2,3]. Hence, in our study CDG participants were also inquired about their perception of the overall severity (considering all the clinical manifestations) of the patient’s CDG. Among PMM2-CDG patients with infection-related issues there was a significantly higher predominance of overall more severe clinical phenotypes (49.2%, *n* = 32/65) (Figure 3I).

### 3.6. Infection Suspicion Is a Major Medical Care Trigger With Antibiotics Reported as the Most Common and Effective Infection Treatment in PMM2-CDG

Participants were inquired about their first attitudes towards infection suspicion. The most named first action by all participants was discussing the possibility of infection with the general practitioner (GP)/pediatrician (56.9% of PMM2-CDG and 56.6% of control) (Appendix A). As for other actions, on the one hand, PMM2-CDG participants were more likely to immediately go to the hospital (18.4%, *n* = 12/65) or to contact their CDG/immunology specialist (7.7%, *n* = 5/65). On the other hand, PMM2-CDG participants showed a slightly lower tendency to self-medication (13.9%, *n* = 9/65) than controls (17.00%, *n* = 9/53) (Appendix A). In non-PMM2-CDG participants, the tendencies were inverted, i.e., they showed an augmented trend of self-medication (24.6%, *n* = 14/57) and a lower likelihood to consult their GP/pediatrician (49.1%, *n* = 28/57) or visit the hospital (8.8%, *n* = 5/57) (Appendix A). These results are in accordance with the increased reports of severe infections and/or infections as triggers to other clinical manifestations made by PMM2-CDG patients (Figure 3) as well as with higher hospitalization rates related to infections observed in these patients (Figure 3C,D).

The use and perceived effectiveness of some infection treatments were investigated in our study. Antibiotic use was high both in PMM2-CDG (96.9%, *n* = 63/65) and control participants (96.2%, *n* = 51/53) (Appendix A). Conversely, intravenous immunoglobulin infusions (IVIG) and immunomodulators were rarely used. The former was more used in PMM2-CDG (12.3%, *n* = 8/65) and the latter by the controls (11.3%, *n* = 6/53). Compared to antibiotics (21.56% of the control, n = 11/51—and 17.45% of PMM2-CDG participants—n = 11/63), their perceived ineffectiveness was higher (Appendix A). The use and perceived effectiveness of antibiotics and IVIG in PMM2-CDG is in accordance with literature data [13]. Information on the use and effectiveness of antivirals is lacking and it has not been addressed in our study either. Additionally, no reports of stem cell/bone marrow transplantation to treat infection-related issues were made by any participants. In the future, it would be interesting to assess the use of antivirals in PMM2-CDG (and other CDG), as it could give hints about causal agents.

### 3.7. PMM2-CDG Have Low Overall Allergy Prevalence but a Significant Presence of Food Allergies

There are a few reports on allergies in CDG, including rhinitis, food and drug allergies [14,16]. To further elucidate their types and prevalence, participants were inquired about specific allergy diagnoses.

Total allergy prevalence was significantly lower in PMM2-CDG patients (33%, *n* = 41/122) (Appendix A). Interestingly, 46.3% of PMM2-CDG (*n* = 19/41) and 52.5% of control participants (*n* = 83/158) reported multiple allergies (≥2). In PMM2-CDG the pattern of allergies was generally more balanced than in the control, with a particularly lower number of sinusitis (9.8%, *n* = 4/41) and rhinitis (17.1%, *n* = 7/41), and a significantly higher food allergy occurrence (26.8%, n = 11/41) (Figure 4A–D). In non-PMM2-CDG patients total allergy prevalence was of 43.7% (*n* = 38/87), closer to but still inferior to the controls (45.3%, *n* = 158/349) (Appendix A).Allergy presence was not associated with increased overall phenotypic severity.

Age was not found to have a distinct effect on allergies. Nonetheless, PMM2-CDG patients displayed a greater tendency to attribute to age a positive effect on allergies (51.2%, *n* = 21/41) than controls (43.7%, *n* = 69/158) (Appendix A). Studies have shown an interplay between allergies and infections [44,45]. Additionally, PMM2-CDG patients (29.3%, *n* = 24/82) showed a higher relative prevalence of concomitant allergies and infections when compared to the control (15.9%, *n* = 29/182) (Figure 4E). Nevertheless, only 9.8% of PMM2-CDG (*n* = 4/41) and 7% of control participants (*n* = 11/158) reported any known association between allergic episodes and infections.

### 3.8. Infections and Allergies Associate with Relevant and Specific Non-Immune Phenotypic Features

CDG mostly show multi-system involvement [2,3]. Hence, a list of clinical signs and symptoms common across CDG and/or which could have a (causal or explanatory) relationship with immunological issues was created. Participants were asked to select the manifestations the patients had been diagnosed with, in order to investigate a possible link between immune-related manifestations and other clinical features. Infections in PMM2-CDG were associated with heart, neurologic and hepato-gastrointestinal manifestations. Indeed, some of these associations overlapped with severe overall phenotypes (Figure 5, Appendix A). This is not surprising, since there was a higher prevalence of severe phenotypes among PMM2-CDG patients reporting infections (Figure 3I). However, some manifestations were exclusive of patients with relevant infections, such as SLE (*p* = 0.00012/OR = 5.67), seizures (*p* = 0.00027/OR = 4.50) and high transaminases (*p* = 0.011/OR = 2.68). Other clinical features only significantly correlated with severe phenotypes, including thrombocytopenia (*p* = 0.0019/OR = 6.39), protein-losing enteropathy(*p* = 0.019/OR = 4.80), proteinuria (*p* = 0.0056/OR = 3.86) and nephrotic syndrome(*p* = 0.014/OR = 5.50). Regarding allergies, only osteopenia/osteoporosis (*p* = 0.0010/OR = 4.08) showed a strong correlation (Appendix A). Patients who reported not having any of the listed clinical signs (*n* = 4) had milder CDG phenotypes.

### 3.9. Infections and Allergies Negatively Impact the Daily Lives of PMM2-CDG Patients

Health-related QoL has been gaining research momentum and clinical importance, also in inherited metabolic diseases [46]. This is a subjective measure which only the people living with (or caring for patients with) the disease can provide. Other studies have shown that the combination of hard clinical and subjective measures, like QoL, might improve disease management and treatment [21,47]. The impact of infections and allergies on the participants’ QoL and daily activities was examined in our study to determine the burden of immune manifestations. An important percentage of participants defined the impact of infections on overall QoL as negative or as the most impactful health-related factor (43.1%, *n* = 28/65 of PMM2-CDG and 37.73%, *n* = 18/53 of control participants). Allergies had a mildly negative impact on QoL (53.7%, *n* = 22/4 PMM2-CDG and the 51.9%, *n* = 82/158, control group) (Figure 6A,B). Some PMM2-CDG (9.2%, *n* = 6/65 of infections and 4.9%, *n* = 2/41 of allergies) and non-PMM2-CDG participants (16.1%, *n* = 9/56 of infections and 2.8%, *n* = 1/36 of allergies) considered that these clinical manifestations worsened other CDG signs and symptoms (Appendix A).

Concerning their impact on daily activities, immune-related signs had a prominently more deleterious impact on PMM2-CDG patients (55.4%, *n* = 36/65 in infections and 29.3%, *n* = 12/41 in allergies) than on the controls (28.3%, *n* = 15/53 in infections and 9.5%, *n* = 15/158 in allergies) (Figure 6C,D and Appendix A). These disparities in the daily impact of immune features might be explained by the differences in their perceived severity, at least regarding infections (50.9%, *n* = 27/53 of controls assessed infections as mild compared to 26.1% of PMM2-CDG participants, *n* = 17/65) (Appendix A). However, in the PMM2-CDG and control groups, increased infection and allergy severity were directly correlated with a higher impact on both general QoL and daily tasks (Table 1). In our dataset, the presence of multiple (≥2) frequent/chronic infections or allergies was not associated with a more negative impact on QoL in the PMM2-CDG and control groups.

### 3.10. No Clear Genotype-(Immune) Phenotype Correlation Identified in PMM2-CDG

Fifty-four percent of the PMM2-CDG participants (*n* = 66/122) shared 36 distinct variants, comprising 39 genotypes. Close to 92% of these variants were missense (*n* = 32/36) or nonsense (*n* = 1/36). Rare variants included an intronic variant causing splicing changes and a duplication resulting in a frameshift variant. As expected, the most common was the p.R141H variant (62.1%, *n* = 41/66), and the most represented genotype was the p.R141H/p.V231M (12.1%, *n* = 8/66). All but one patient—harboring a p.P113L/p.P113L in homozygosity due to a maternal disomy—were compound heterozygotes (Appendix A). Consulted mutation databases had an overall good coverage of PMM2 variants. Even so, two newly reported variants (c. 98A > C, p.Q33P and c.140C > T, p.S47L) were present in one patient (Appendix A). Most in silico pathogenicity prediction tools uniformly ranked the reported variants as deleterious (Appendix A).

A genotype-immune phenotype relationship was studied among those PMM2-CDG participants who shared variants. Our PMM2-CDG group could be divided into patients harboring or not the p.R141H variant (Appendix A). The subset of p.R141H patients presented a higher relevant infection prevalence (56.1% *n* = 23/41). Globally, allergies (36%, *n* = 9/25) were slightly more frequent in patients lacking this variant. However, 6 of the patients who specifically reported food allergies harbored the p.R141H variant. Regarding the most common genotype in our group—the p.R141H/p.V231M—early lethality was reported in three patients and infections were described to be the cause of death in two of them (Appendix A). Similarly, Schiff and collaborators described the premature death of all p.R141H/p.V231M patients in their cohort [15]. On the other end of the spectrum were patients (*n* = 3) with the p.P69S missense variant (classified as benign by most of the in silico prediction tools) with a mild to moderate phenotype without any immune-related problems (Appendix A). Interestingly, the p.P69S was in combination with variants mostly scored and annotated as pathogenic (two patients p.R141H/p.P69S and the one p.D148N/p.P69S) (Appendix A). No significant genotype-immunehenotype or genotype-phenotypic severity associations could be drawn from our data (not shown). Nevertheless, a tendency to have more severe overall phenotypes was seen in p.R141H patients (OR = 1.35). A significant relationship between the p.R141H substitution and proteinuria was also found (*p* = 0.027/OR = 2.91) (Appendix A).

### 3.11. High Vaccination Compliance, with Greater Perceived Ineffectiveness and More Severe Adverse Reactions (ADRs) in PMM2-CDG

Despite some scattered reports of vaccination ineffectiveness in CDG, namely due to the premature loss of or incapability to produce protective antibodies, a comprehensive understanding of vaccination compliance, monitoring, effectiveness and ADRs is lacking [13,14,17]. To bridge this relevant knowledge gap, a section on vaccination history was included in this study.

The vast majority of all study participants have been vaccinated (Appendix A). Indeed, only three PMM2-CDG patients were not vaccinated at all and have suffered from relevant infections. Most participants also have received all the usual vaccinations. Nevertheless, more PMM2-CDG participants (7.8%, *n* = 9/116) did not take all vaccines compared to controls (2.7%, *n* = 9/338) (Figure 7A). To make vaccination-related choices, PMM2-CDG participants tended to mainly rely on medical advice (78.8%, *n* = 93/118) while controls resorted significantly more to nurse recommendations (21.8%, *n* = 76/349) and parent/participant decisions (67.0%, *n* = 234/349) (Figure 7B). Two PMM2-CDG respondents also mentioned they seek other PMM2-CDG families’ advice to make decisions about vaccination. Vaccination-related shared decision-making (e.g., involving both doctor/nurse and parent/participant) was not significantly different between the studied groups.

Vaccination response monitoring is very useful to confirm efficient immunization. However, vaccination monitoring, by the measurement of antibody titers following vaccination, was rarely performed in PMM2-CDG patients (12.7%, *n* = 15/118) and controls (16.2%, *n* = 55/339) (Appendix A). Nevertheless, perceived vaccination ineffectiveness was higher in the PMM2-CDG group (28.3%, *n* = 17/60) (Figure 7C). Among these PMM2-CDG for whom vaccination was perceived to be ineffective, identified reasons for vaccination ineffectiveness included loss of or failure to produce protective antibodies (35.3%, *n* = 6/17), and illness caused by a pathogen against which the patient had been vaccinated (29.4%, *n* = 5/17). In the remaining participants the cause was unknown. As for the reasons described by control participants, either protective antibodies loss/lack of production (*n* = 6/14) or a subsequent infection by the same pathogen against which they had been vaccinated (*n* = 6/14) was reported by 85.7% of participants. Two control participants did not know why vaccination was ineffective.

Adverse reactions or events of varying severity have been documented for most vaccines. Most documented ADRs are mild, such as local swelling and pain, as well as mild fever [48]. The total number of PMM2-CDG participants (55.1%, *n* = 65/118) who reported vaccination-related ADRs was not significantly different from controls (43.9%, *n* = 149/339). However, occurrence of fever was more frequent among PMM2-CDG (52.5%, *n* = 62/118) than among controls (39.5%, *n* = 134/339). Moreover, PMM2-CDG patients were reported to have more severe ADRs, such as generalized infections (3.4%, *n* = 4/118), seizures (*n* = 1/118) and SLE (*n* = 1/118) which were absent in the controls (Figure 7D).

### 3.12. PMM2-CDG Participants Describe Few Immune Cell and Protein Altered Counts but Have Higher Immune Monitoring and Information Needs

Altered immune cell and protein counts have been described in CDG [13,14]. However, these accounts are scarce and often little comprehensive. Hence, to complete existing data and evaluate the knowledge of CDG patients/caregivers, a question focused on laboratory immune counts was included in the questionnaires. As available in the CDG literature, alterations in the counts of immune cells and proteins were seldom reported by participants (Appendix A). Moreover, an important percentage of participants were unaware or uncertain of any alterations ever being detected (20.8% of PMM2-CDG—*n* = 25/120, 22.9% of non-PMM2-CDG—*n* = 19/83, and 7.8% of control participants—*n* = 23/293). Among PMM2-CDG, only a small subset of patients described constant alterations with the most common being leukocytosis, monocytosis and low antibody counts (Appendix A). All but three PMM2-CDG patients who reported constantly altered laboratory counts simultaneously had relevant infections. Noteworthy, PMM2-CDG patients with relevant infections were significantly more likely to have had their immune-related manifestations related to their underlying CDG (Appendix A). However, neither relevant infections nor increased overall severity correlated with an earlier diagnosis in PMM2-CDG, probably due to their clinical unspecificity. Moreover, only 7 PMM2-CDG participants reported being followed by an immunology specialist (Appendix A).

Literature data lacks information about immune testing, namely about types of tests and testing periodicity. Additionally, it does not provide any insight on the immune-related information needs of CDG patients/caregivers. To create knowledge in unexplored topics and possibly identify new unmet needs, some questions were dedicated to these subjects. Regarding immune parameters monitoring, 71.3% of PMM2-CDG patients (*n* = 87/122) were reported to have done tests for this purpose and only 10.7% (*n* = 13/122) of the participants said no immune monitoring tests had been done. Among control participants, 61% (*n* = 213/349) reported immune testing but over 32% (*n* = 113/349) reported never having done any type of tests to monitor immune parameters/function. In the non-PMM2-CDG group, 65.5% of the participants (*n* = 57/87) identified at least one immune-related test, while 11.5% (*n* = 10/87) reported never having done any. Across all groups, the most named test was the complete blood count (CBC) (Figure 8A and Appendix A). Focusing on specific tests, compared to controls, significantly more PMM2-CDG participants mentioned C-reactive protein (CRP) measurements (43.4%, *n* = 53/122) (Figure 8A). This higher number of participants recognizing CRP measurements could be associated with the pronounced and/or frequently found CRP alterations during infection or other clinical problems. Controls identified more allergy monitoring tests, such as skin prick (18.6%, *n* = 65/349), breath (9.2%, *n* = 32/349) and delayed-type hypersensitivity skin testing (4.3%, *n* = 15/349). Interestingly, these patterns of immune tests identified by PMM2-CDG and control participants are concordant with the most frequent types of immune-related manifestations described by both groups. No other remarkable differences were found regarding the different tests performed, either for PMM2-CDG and control participants or between participants with or without immune-related manifestations. In terms of test frequency, 71.4% PMM2-CDG participants (*n* = 35/49) and 72% of non-PMM2-CDG participants (*n* = 27/37) reported more frequent testing, while 58.9% of controls (*n* = 43/73) reported doing immune tests only once (Figure 8B and Appendix A).

Concerning information needs of immunology-related topics, most participants expressed their interest in getting more information without any statistically significant differences related to presence or absence of immunological involvement (Appendix A). Nonetheless, PMM2-CDG participants (83.6%, *n* = 102/122), as well non-PMM2-CDG patients (78.2%, *n* = 68/87), requested significantly more information on the topic when compared to the control group (50.6%, *n* = 176/348) (Appendix A).

### 3.13. Participants Reported Good Content Understandability, Management, and Glossary Usefulness

At the end of the questionnaires three questions about the content understandability, use and usefulness of the glossaries provided, and management experience were included. This was done to further assess content appropriateness and as an accessory measure to validate the quality of the information provided by participants [49]. Most respondents reported a complete understanding of the content, but the overall understanding of the questions was significantly better in the control group (84.1%, *n* = 276/328, Appendix A). Regarding glossary use, PMM2-CDG respondents resorted more to glossaries during the completion of the questionnaire (58.6%, *n* = 68/116), but they (66.2%, *n* = 45/68) did not find them more useful than other respondents (78.8%—*n* = 115/146—of control and 65%—28/43—of non-PMM2-CDG participants) (Appendix A). Still, most participants described glossaries as helpful. As for experience management, most participants felt they managed well on their own and would not need help from a third party. Nevertheless, both PMM2-CDG (11.3%, *n* = 13/115) and non-PMM2-CDG participants (16.5%, *n* = 14/85) felt significantly more difficulties in filling out the questionnaire without help than controls (3.7%, *n* = 12/328) (Appendix A). Detailed descriptions of understandability, management and glossary use are depicted for the control (Appendix A), PMM2-CDG (Appendix A) and non-PMM2-CDG groups (Appendix A). These greater difficulties are likely to stem from the medical complexity of CDG and the higher information needs of this community. All in all, these results emphasize the questionnaires’ and other materials’ appropriateness, but also the readiness of participants to partake in this study.

## 4. Discussion

Immunological involvement in CDG is an underexplored field. Even though the ubiquitous importance of glycans in the immune system and response is well-established, clear molecular and clinical evidence capable of fully clarifying the immune phenotype of CDG patients, particularly in PMM2-CDG, is lacking [13,14,16,17,50,51,52]. This lack of knowledge regarding immunological involvement is evidenced in this study by: (i) the ~11% of PMM2-CDG patients with unsolved immune status and (ii) the uncertainty or ignorance of PMM2-CDG participants about the presence of specific immune-related manifestations (6.6%, 7.4% and 16.4% related to allergies, infections and autoimmune diseases, respectively).

Our study confirmed previously reported infection-related features in PMM2-CDG, namely (1) the higher prevalence of infections during infancy/early childhood; (2) their organ-specific pattern (particularly RT and GI); (3) the not negligible occurrence of sepsis evolving from organ-specific infections and iv) their associated early lethality [13,14,16,17,52,53]. All these observations highlight the need for close monitoring until full remission occurs to ensure recovery and to prevent clinical aggravation and fatal outcomes [17]. Additionally, this study provides de novo knowledge of a complex and diverse infection pattern with a noteworthy prevalence of GI infections often identified as a reason for hospitalization. Additionally, a high clinical impact of infections as triggers to other clinical manifestations such as SLE and seizures was found [15,18]. Indeed, SLE and seizures have been concomitantly reported in PMM2-CDG [15]. A proinflammatory immune response has been postulated as a core epilepsy cause, and the sometimes successful use of anti-inflammatory drugs (e.g., corticosteroids) to treat epilepsy supports this assumption [54,55]. The most frequently used anti-epileptics in PMM2-CDG, carbamazepine, levetiracetam and valproic acid, have anti-inflammatory properties [56,57,58]. These are also recommended as therapeutic agents during SLEs [13].

PMM2-CDG patients can be divided into those with a neurological and those with a neurovisceral phenotype [42]. Besides infections associated with specific neurological signs, our results suggest that they are also significantly associated with hallmarks of the neurovisceral phenotype, such as heart (pericardial effusion and cardiomyopathy), liver (high serum transaminases) and GI involvement (chronic diarrhea, feeding tube and gastroesophageal reflux) [15]. Previous studies have linked pericardial effusion, hypertrophic cardiomyopathy and inflammatory or infectious issues and have documented an increase of serum transaminases upon viral infections in PMM2-CDG [13,59,60].

The strong association between infections and GI manifestations in our PMM2-CDG group could stem from the high prevalence of GI infections in our patients. Nevertheless, Kjaergaard and collaborators observed that GI signs such as vomiting and diarrhea were exacerbated by intercurrent infections [61], suggesting that a more complex relationship between infections and GI manifestations could exist.

Additionally, a correlation between infections and increased overall phenotypic severity was detected in our PMM2-CDG group. This accentuates the theory that infections are connected to the neurovisceral phenotype. In addition, overall phenotypic severity was related to protein-losing enteropathy, renal problems (proteinuria and nephrotic syndrome), and thrombocytopenia in our PMM2-CDG dataset. Renal manifestations are reported to be more frequently present in the severe visceral PMM2-CDG, although mild proteinuria is a frequent manifestation of CDG [62]. That thrombocytopenia is significantly related to severe phenotypes but not with infections, might indicate that it is not of immune origin in PMM2-CDG [60].

Apart from the association of the p.R141H/p.F119L genotype and the p.R141H variant with increased phenotypic severity, and the more recent suggestion of milder PMM2-CDG associated with the p.L32R variant, there are no strong genotype-phenotype relationships in PMM2-CDG [61,63]. Its clinical severity is probably also determined by the presence of variants in other (glycosylation) genes [64,65]. This complicates a direct association between a known variant or genotype and a set of clinical features. In this study, variable genotypes were recorded without any concrete found association between a genotype or a single variant and immune-related manifestations. In our PMM2-CDG group, as in the literature, the p.R141H variant is the most common. Even though a relationship between this mutation and phenotypic severity was not found, it was significantly associated with proteinuria [12,61]. To better unveil possible genotype/variant—(immune) phenotype relations in PMM2-CDG and other CDG, a larger patient sample is needed.

Glycosylation is fundamental for pathogen recognition and internalization, and glycosylation defects can favor either pathogen resistance or susceptibility, namely by the activation of compensatory or parallel pathways. For instance, accumulation of mannose-6-phosphate (M6P)—which occurs when PMM2 activity is impaired—has been shown to instigate lipid-linked oligosaccharides (LLO) destruction upon herpes simplex virus infection. M6P-mediated LLO depletion affects host N-glycosylation and, in turn, impedes viral infection and propagation [66]. Another textbook example is MOGS-CDG: patients show an increased resistance to enveloped virus, as a result of impaired host N-glycosylation [67]. Concordantly, the three MOGS-CDG participants in our study did not identify virus as the most common infectious agent. Further, a greater susceptibility to a type of microorganism can be connected to specific underlying immune defects. For example, a defective innate immune (what happens in SLC35C1-CDG) or humoral response is associated with bacterial infections, while T-cell defects underlie viral and fungal infection susceptibility [68,69]. Additionally, generalized immunodeficiency phenotypes (e.g., PGM3-CDG) can also be linked to opportunist pathogens [70]. This variety of somewhat counterbalancing mechanisms may help explain the lack of a clear infectious agent pattern in our CDG dataset. Nonetheless, it also highlights the need for a more systematic and detailed study of CDG infectious agents.

Allergen glycan make-up has been shown to play an important role in their recognition and uptake whereas the impact of host altered glycosylation has remained more elusive [71]. However, a recent study has shown that defective IgE glycosylation can decrease allergic responses [72]. Indeed, in our dataset, allergies were significantly lower among PMM2-CDG participants; the only exception were food allergies which were significantly more prevalent in these patients. These trends are in accordance with the reported organ-specific infections, which might help explain organ-specific immune dysfunction [45]. Much atopy was present among PMM2-CDG participants with a highlight for the co-occurrence of food allergies and eczema or dermatitis (*n* = 5/11). Indeed, a relationship between atopic dermatitis or eczema and food allergies has been reported in pediatric patients [73]. Interestingly, in PMM2-CDG, allergies significantly co-segregated with osteopenia/osteoporosis. Low bone density is a recognized but not well understood PMM2-CDG manifestation [37]. Recent studies have hinted to a link between gut microbiota and osteoporosis, and between bone manifestations and immune problems [74,75]. However, certain immunomodulators, such as corticoids, can also impact bone density [76,77]. Among non-PMM2-CDG, skin allergies—particularly eczema—were the most reported, especially in ALG13-, SRD5A3- and PIGA-CDG. Noteworthy, the *Piga* null mouse model develops the Harlequin ichthyosis phenotype due to impaired processing of filaggrin, a protein involved in epidermal cell permeability and homeostasis [78]. This is associated with ichthyosis and atopic eczema [79] and could help account for its presence in PIGA-CDG.

Autoimmune disorders result from loss of immunologic (self-)tolerance. Altered cellular and humoral responses may unleash an autoreactive response with glycans as important determinants [80,81]. Nevertheless, autoimmunity had a very low prevalence in our CDG groups, with confirmed diagnoses in only three PMM2-CDG patients, all affecting the GI tract and including celiac and inflammatory bowel diseases. Of note, screening for protein/gluten intolerance disorders has been added to the clinical guidelines [13], highlighting the need to better clarify the prevalence and etiology of these manifestations in PMM2-CDG. Besides PMM2-CDG, three reports of autoimmune diseases have been associated with the ALG gene family (eosinophilic esophagitis in an ALG1-CDG patient with a severe phenotype, and psoriasis in an ALG3-CDG and an ALG6-CDG patient). Celiac disease was reported in a MAN1B1-CDG patient with a remarkable GI phenotype. Hemolytic anemia was reported in a PIGN-CDG patient which may stem from GPI-anchor deficiency of red blood cells. GPI-anchors regulate the erythrocyte-stimulatory/lytic cycle, and hemolytic anemia is associated with paroxysmal nocturnal hemoglobinuria, caused by somatic *PIGA* variants [82].

Despite similar reported autoimmunity testing (11.5%, *n* = 24/209, in CDG and 9.2%, *n* = 32/349, in controls), the suspicion of autoimmune diseases was much higher in the CDG group (11.5%, *n* = 24/209 in CDG and 4.6%, *n* = 16/349 in the controls). This could be attributed, at least in part, to the lack of explanation for some of the manifestations shown by the CDG patients and/or by the fact that some of the proposed autoimmune diseases (e.g., arthritis or eosinophilic esophagitis) have overlapping symptoms with non-autoimmune manifestations. Nevertheless, the generally low prevalence of autoimmune diagnoses could also result from the CDG represented in our group which lacks (e.g., G6PC3-, SLC37A4- and GALNT3-CDG) more extensive autoimmunity reports [14].

PMM2-CDG comorbidities and specific medication intake may justify the observed higher vaccination selectiveness and perceived ineffectiveness [83]. Although inadequate vaccination responses have been reported in PMM2-CDG in this and other studies [17], age-appropriate vaccination is recommended, and re-vaccination should be carefully considered. Regarding vaccination ADRs, only fever was significantly more frequent in PMM2-CDG. Of note, a recent case-report suggested that a post-vaccination fever in a PMM2-CDG patient harboring the p.V231M variant aggravated the phenotype, resulting in the patient’s death [84]. The thermolability of this variant was suggested to have contributed to this outcome [85]. Noteworthy, severe vaccine ADRs in PMM2-CDG patients included generalized infections, SLE and seizures in a small patient subset. Hence, our study also emphasizes the need for a more systematic monitoring of vaccination effectiveness, detailed vaccination response and ADRs to better inform HCPs and families’ choices.

Both in the literature and our study, PMM2-CDG patients presenting relevant abnormal immunological laboratory counts constitute a small group [13,14]. Nevertheless, the constant alterations noted in our study are suggestive of a proinflammatory response. Preliminary unpublished data from our group has shown that PMM2-CDG patient T-cells show higher proliferation rates and IFN-γ production upon mitogen stimulation. Maybe PMM2-CDG patients are in a permanent inflammatory state because of the higher expression of sialyl Lewis^x^ on α_1_-acid glycoproteins [51]. Substantiating this persistent proinflammatory state are the reports of FUFEs [13,43], particularly prevalent in our study among PMM2-CDG patients with infections. However, while most PMM2-CDG participants with permanent alterations also reported relevant infections, this fact alone could be responsible for these constantly elevated counts. More systematic immune testing and data collection are required to better understand the prevalence, evolution, and medical relevance of these alterations as well as to help establish a concrete diagnosis/monitoring strategy. Better clinician-family communication and literacy on this topic is pivotal since a high number of participants were unaware or incapable of answering about laboratory counts. On the one hand, glycan anomalies in immune proteins and cells may primarily contribute to defective or reduced activation/function, as opposed to altered counts [13,86]. On the other hand, low expression/counts of immune proteins or cells may not have relevant clinical consequences [68]. These facts must be clearly elucidated and communicated. Participants expressed interest in receiving more information on immunology which underlines an unmet information demand concordant with a previous study conducted in the CDG community [22,87,88,89]. Health education and literacy for non-experts are becoming increasingly important, particularly as we are entering an era of personalized medicine that will require patients to play an active role in health decision-making [90]. Having patients/families as research partners—and not only mere participants—is the logical next step in this new paradigm [91].

### 4.1. Limitations of the Study

-Possible sampling bias created by (a) overrepresentation of social media users and/or of people involved in health/science-related associations, and (b) increased likelihood of people experiencing immune-related disturbances participating. Actions to minimize these possibilities include: (i) developing an informative and multi-platform recruitment strategy; (ii) adding logic to specific questions which result in a longer questionnaire if immune issues were present.-Gender and adult prevalence differences between the PMM2-CDG and controls. However, random subsampling using the sample () R function was attempted and no significant differences were found. Moreover, many tests were done between sub-groups (e.g., participants with relevant infections) and not through the comparison of the complete samples. Detailed descriptive characterization of these sub-groups is always provided.-Age ranges implementation, especially without the definition of a maximum age limit in adults, can lead to loss of information and non-optimal group characterization. However, within the subsections of the questionnaires, we added questions that allowed for an age/time framing to help overcome this limitation. Furthermore, in the open question format, participants often did not specify whether they were referring to age in months or years, resulting in significant data loss [21].-Although the ImmunoCDGQ was built with the intent to capture data from all CDG, it was mainly focused on PMM2-CDG, by far the most frequent CDG.-A more comprehensive listing of the types of medications taken by CDG patients is needed to properly establish their contribution to the onset, aggravation or improvement of clinical signs and symptoms, especially of those related to the immune system.

Although the questionnaires could benefit from some structural and content refinements, lacking further validation, most participants reported a good understanding of the tool and felt comfortable replying on their own.

### 4.2. Strengths of the Study

-Creating a dataset of 209 CDG patients including 122 PMM2-CDG patients. To date, approximately 1000 PMM2-CDG patients have been reported and even in large cohort studies, no such large number of patients has been gathered before [15,42].-In a short period of time, our approach allowed the collection of data on a highly neglected medical and scientific topic in CDG, allowing the generation of new data and knowledge in a time- and cost-efficient manner.-This project paves the way for further progress and validation of people/patient-centered research. It also shows that citizens, particularly rare disease patients and caregivers, are willing to be involved in research and share their unique knowledge.-This study also provided new data on QoL, highlighting the negative impact of immune-related manifestations—particularly of infections—on the daily life of the patients. This information helps to better understand the full impact of immune signs and symptoms in CDG.-An informative and education campaign on immunology (and CDG) preceded data collection. Moreover, many support measures and resources were developed to both stimulate participation and improve the understanding of the content. Hence, this study also had an impact on health literacy.-Our results point towards a new research avenue: the need for exploring GI (dys)function, more particularly the microbiome of PMM2-CDG patients since PMM2-CDG GI clinical signs were associated with all immune manifestations evaluated in this study. The role of the gut, most specifically of the microbiome, in the immune response and metabolism is just beginning to be uncovered, and our findings clearly indicate that PMM2-CDG pathophysiological comprehension could benefit from studies on this topic [92,93,94].

## 5. Conclusions

Immunological involvement in CDG is complex, and this hinders the pathophysiological elucidation of immune-related manifestations. It remains to be determined if these immune-related clinical manifestations and/or laboratory alterations directly result from glycan deficiencies, other clinical features (e.g., gastroesophageal reflux can favor RT infections, proteinuria can cause Ig loss and bone marrow failure can lead to neutropenia), specific medications, or from a combination of these variables. This study has shed light on immune-related manifestations across 35 CDG, mainly PMM2-CDG.

In PMM2-CDG, we could disclose an axis formed by the neurologic, GI, and immune systems. Better understanding of the immune system-gut crosstalk in PMM2-CDG can open new and important research avenues. Indeed, a systematic study of PMM2-CDG microbiome (in terms of content, glycosylation status and metabolome) could be interesting, to identify possible prognostic and therapeutic biomarkers. Use of probiotics and the performance of fecal microbiota transplants and their effectiveness should be studied in PMM2-CDG patients. Moreover, this work has raised flag marks on issues that require better elucidation and monitoring to improve clinical management. Our results show that immune-related manifestations, particularly infections, represent a substantial burden to PMM2-CDG patients, their families and, ultimately, healthcare services. Therefore, reliable management and empowerment strategies—for professionals and CDG caregivers/patients —should be put in place to help the CDG community better deal with these issues.

## Figures and Tables

**Figure 1 jcm-09-02092-f001:**
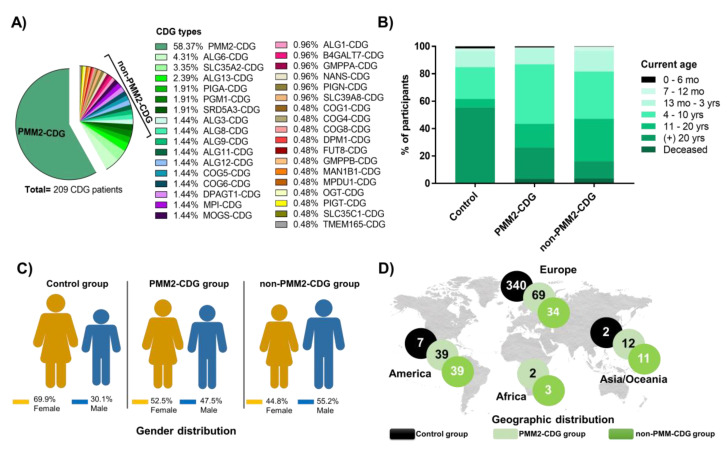
Characteristics of the ImmunoCDGQ (PMM2-CDG and non-PMM2-CDG groups) and ImmunoHealthyQ (control group) participants. (**A**) CDG distribution. CDG are listed in descending order according to their prevalence in the ImmunoCDGQ; (**B**) Age distribution of participants at the time of participation in the study; (**C**) Gender distribution of the study participants. Females are represented in yellow and males in blue; (**D**) Geographic distribution of participant per continent.

**Figure 2 jcm-09-02092-f002:**
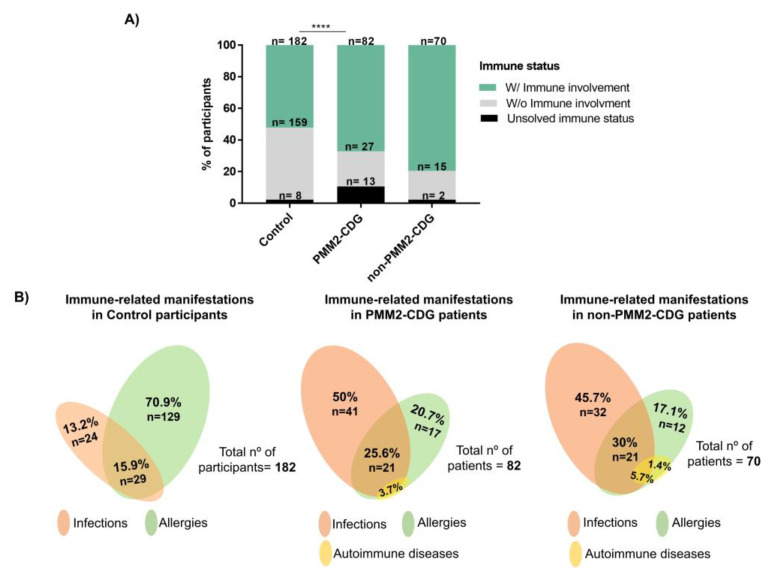
Prevalence and types of immune-related clinical features. (**A**) Overall immunological involvement. Three categories based on the immune manifestations reported were established. Statistical significance refers to the comparison between the prevalence of participants with immunological involvement (reporting, at least one clinical immune feature) and without immunological involvement (not reporting any immune-related manifestations) in the control and PMM2-CDG groups (*p* = 5.19 × 10^−5^(****)/OR = 2.65). Participants with unclear immune phenotype were assigned to the unsolved immune status category; (**B**) Types of immune-related manifestations (infections, allergies, and autoimmune diseases) reported by participants assigned to the immunological involvement category. The overlapping region of the Venn diagrams represents the concomitant presence of two or three immune manifestations. All statistical significance was determined by Fisher’s exact test.

**Figure 3 jcm-09-02092-f003:**
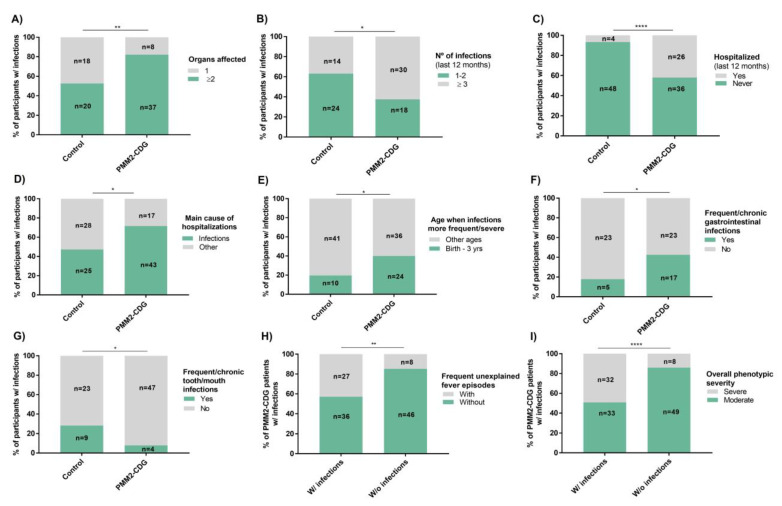
Multiple infections, associated hospitalizations, frequent unexplained fever episodes (FUFEs) and overall increased phenotypic severity. (**A**) Presence of multiple infections (affecting different organs) at present. Statistical significance refers to the comparison between participants reporting infections in at least two different organs (grouped) and participants reporting infection in only one organ (*p* = 0.0047 (**)/OR = 4.09) Dead patients were excluded from this analysis; (**B**) Nº of infections in the past 12 months. Respondents who replied 3–6, 7–10 and 10+ infections were grouped for this analysis. Statistical significance refers to the comparison between participants reporting 1–2 and patients reporting >3 infections (*p* = 0.029 (*)/OR = 2.82); (**C**) Infection-driven hospitalizations in the past 12 months. Participants who replied, “I don’t know” and dead patients were excluded from this analysis. Statistical significance refers to the comparison between participants who had experienced infection-related hospitalizations and those who had not (*p* = 3.22 × 10^−5^ (****)/OR = 8.51); (**D**) Infections as the main cause of hospitalization. Participants who replied that infections were or had been the main cause of hospitalization were grouped forming the “Infections” group, while those who replied, “I don’t know” and dead patients were excluded from this analysis. Statistical significance refers to the comparison between participants identifying infections as the main cause of hospitalization against those who did not (*p* = 0.012 (*)/OR = 2.81); (**E**) Age when infections were more frequent and/or severe. Respondents answering infections were more clinically relevant between birth and 3 years were compared to all other participants who reported a different age range (named “Other ages”). Statistical significance refers to the comparison between these two groups of participants (*p* = 0.023 (*)/OR = 2.71). Participants replying “I don’t know” were excluded from this analysis; The most and least prevalent frequent/chronic infections in PMM2-CDG were (**F**) GI infections (*p* = 0.039 (*)/OR = 3.34) and (**G**) frequent/chronic tooth/mouth infections (*p* = 0.026 (*)/OR = 0.22) in controls. Statistical significance refers to the comparison between participants with frequent or chronic infections (grouped) and participants who never experienced these infections; (**H**) FUFEs prevalence among PMM2-CDG patients with and without infections. Statistical significance refers to the comparison between these patients (*p* = 0.0011 (**)/OR = 4.26). Participants unaware of FUFEs presence/absence were omitted from this analysis; (**I**) Overall phenotypic severity in PMM2-CDG patients according to infection presence/absence. Patients with either mild or moderate phenotypes were grouped forming the “Moderate” variable, and patients with severe or very severe phenotypes were combined constituting the “Severe” group. Statistical significance refers to the comparison of the prevalence of “Moderate” and “Severe” phenotypes in PMM2-CDG patients with or without infections (*p* = 3.93 × 10^−5^ (****)/OR = 5.85). All statistical significance was calculated with Fisher’s exact test.

**Figure 4 jcm-09-02092-f004:**
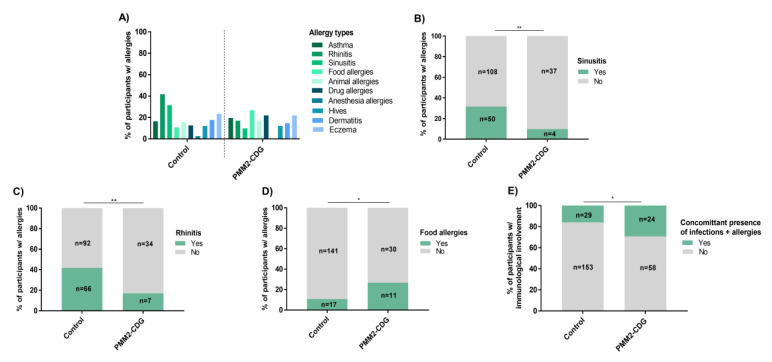
Allergy prevalence. (**A**) Prevalence of specific allergies in PMM2-CDG and control participants. Statistical significance refers to the comparison of total allergy prevalence in PMM2-CDG against the control (*p* = 0.026/OR = 0.61); Prevalence of specific allergies in the PMM2-CDG and control groups, namely (**B**) food allergies (*p* = 0.020 (*)/OR = 3.02), (**C**) rhinitis (*p* = 0.0034 (**)/OR = 0.289) and (**D**) sinusitis (*p* = 0.0052 (**)/OR = 0.23). Statistical significance refers to the comparison between participants reporting rhinitis, sinusitis, or food allergies and those not reporting these allergies; (**E**) Simultaneous presence of allergies and infections. Statistical significance refers to the comparison between the prevalence of participants with and without simultaneous allergies and infections in the control and PMM2-CDG groups (*p* = 0.019 (*)/OR = 2.18). All statistical significance was calculated with Fisher’s exact test.

**Figure 5 jcm-09-02092-f005:**
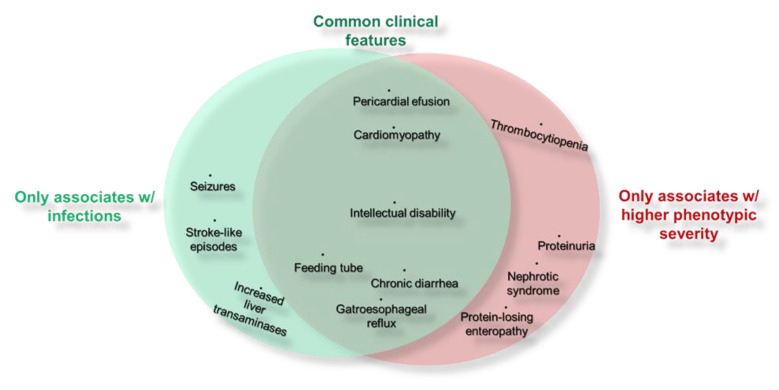
Association between infections, overall phenotype severity and other PMM2-CDG clinical features. Infections and increased overall phenotypic severity significantly correlated with a set of common clinical signs (represented in the middle part of the figure). However, there are clinical features which only significantly segregated with infections (seizures, SLE and increased liver transaminases) or increased phenotypic severity (thrombocytopenia, nephrotic syndrome, and protein-losing enteropathy). Statistical significance calculated with Fisher’s exact test. All *p* values and OR for each clinical manifestation are available in Appendix A.

**Figure 6 jcm-09-02092-f006:**
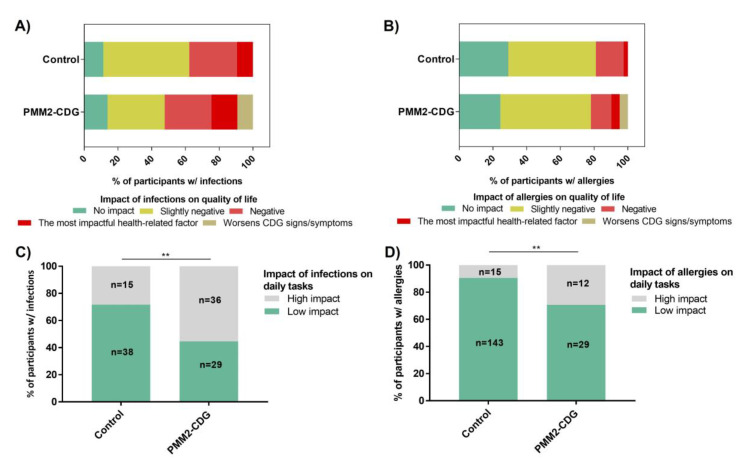
Infections and allergies’ impact on quality of life (QoL). (**A**) Impact of infections on general control and PMM2-CDG participants QoL; (**B**) Impact of allergies on general QoL; (**C**) Comparison of the impact of infections on everyday tasks in the control vs. PMM2-CDG. The options “No impact” and “Slightly negative” were combined forming the “Low impact” group and the same was done for “Negative” and “The most impactful health-related factor” constituting the “High impact” group. Statistical significance refers to the comparison between these two groups, the PMM2-CDG and controls (*p* = 0.0049 (**)/OR = 3.11); (**D**) Comparison of the impact of allergies on everyday tasks in the control vs. PMM2-CDG. “No impact” and “Slightly negative” were combined forming the “Low impact” group and the same was done for “Negative” and “The most impactful health-related factor” constituting the “High impact” group. Statistical significance refers to the comparison between these two groups the PMM2-CDG and controls (*p* = 0.0034 (**)/OR = 3.91). Statistical significance was determined using Fisher’s Exact test.

**Figure 7 jcm-09-02092-f007:**
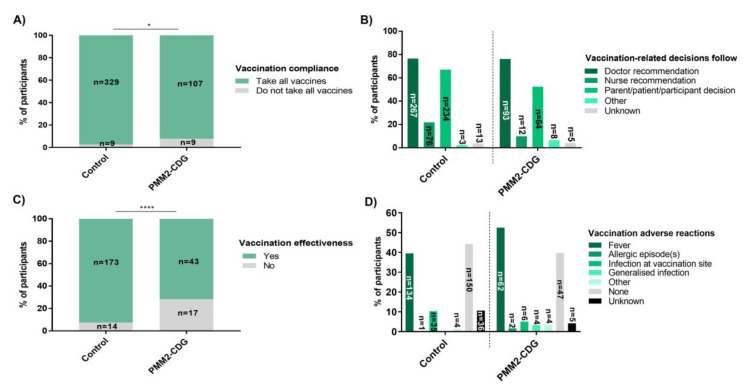
Vaccination compliance, effectiveness, safety and decision-making. (**A**) Vaccination compliance. Participants were asked if they had taken all vaccines available to them. Statistical significance refers to the comparison between participants who reported having taken all vaccines and those who said they had not in PMM2-CDG vs. controls (*p* = 0.024 (*)/OR = 3.07). Participants unaware or uncertain of having taken all vaccines were excluded from this analysis; (**B**) Players involved in vaccination-related decision-making. When compared to the controls, PMM2-CDG participants relied significantly less on nurses’ recommendation (*p* = 0.0029/OR = 0.39) and on parent/participant judgement (*p* = 0.0046/OR = 0.54). Statistical significance refers to the comparison between participants who reported following nurses’ recommendations or parent/participant judgement against those who did not; (**C**) Vaccination perceived (in)effectiveness. Statistical significance refers to the comparison between participants assessing vaccination as effective and those describing it as ineffective (*p* = 8.339 × 10^−5^ (****)/OR = 4.85). Participants who were unaware of or uncertain about the (in)effectiveness of vaccination were omitted from this analysis; (**D**) Specific vaccination adverse reactions (ADRs). In total, PMM2-CDG did not report significantly more vaccination-associated ADRs, except for fever (*p* = 0.017/OR = 1.69). Statistical significance refers to the comparison between participants identifying vaccination-triggered fever and those who did not. In the “Other” category for controls were included edema at the vaccination site (*n* = 2/339), muscular pain with tetanus associated with a yellow fever vaccine (*n* = 1/339) and faintness and transient vision/hearing loss (*n* = 1/339) after every dose of human papilloma virus (HPV) vaccine. For PMM2-CDG in the “Other” group, seizures (*n* = 1/118), stroke-like episodes (*n* = 1/118) and, two patients were reported to have become extremely upset, limp and required medical help. All statistical significance was calculated with Fisher’s exact test.

**Figure 8 jcm-09-02092-f008:**
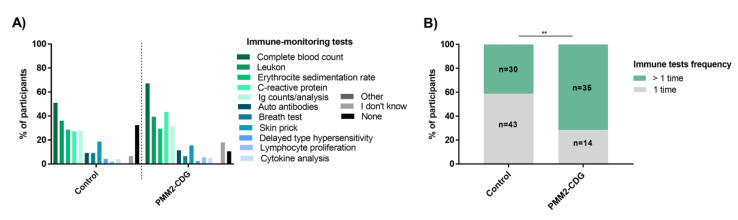
Immune testing. (**A**) Types of immune tests reported by PMM2-CDG and control participants. Only C-reactive protein (CRP) is significantly different between PMM2-CDG and control participants (*p* = 0.0043/OR = 1.87). Statistical significance refers to the comparison between the participants who identified CRP testing and those who did not in the PMM2-CDG and control groups; (**B**) Immune testing frequency. Statistical significance refers to the comparison of participants reporting to have done immune tests once to others reporting having done the tests with greater periodicity, at least twice, which were combined into the “> Once” group (*p* = 0.0015 (**)/OR = 3.54). Participants unaware or uncertain of the frequency of testing were excluded from this analysis. Only participants reporting the presence of immune-related manifestations (prior to this question) and capable of communicating the frequency of testing had to reply to this question. Statistical significance was calculated using Fisher’s exact test.

**Table 1 jcm-09-02092-t001:** Correlation between perceived infection/allergy severity and quality of life (QoL) impact on PMM2-CDG and control groups. The options “No impact” and “Slightly negative” were combined, forming the “Low impact” group. The same was done for “Negative” and “The most impactful health-related factor” constituting the “High impact” group. Statistical significance was determined with Fisher’s exact test. *p*-values and OR are presented.

		Control						PMM2-CDG					
		Impact QoL		*p*-Value/OR	Impact Everyday Tasks		*p*-Value/OR	Impact QoL		*p*-Value/OR	Impact Everyday Tasks		*p*-Value/OR
		High	Low		High	Low		High	Low		High	Low	
**Infection severity**	Mild	14.81% (*n* = 4)	85.19% (*n* = 23)	0.00048/9.69	11.11% (*n* = 3)	88.89% (*n* = 24)	0.0053/7.09	11.76% (*n* = 2)	88.24% (*n* = 15)	6.425 × 10^−5^/18.24	35.29% (*n* = 6)	64.71% (*n* = 11)	0.017/4.45
	Severe	64% (*n* = 16)	36% (*n* = 9)		48% (*n* = 12)	52% (*n* = 13)		72.22% (*n* = 26) ^1^	27.78% (*n* = 10)		71.43% (*n* = 30)	28.57% (*n* = 12)	
**Allergy severity**	Mild	9% (*n* = 9)	91% (*n* = 91)	1.99 × 10^−5^/6.75	2% (*n* = 2)	98% (*n* = 98)	2.05 × 10^−5^/16.44	5% (*n* = 1)	95% (*n* = 19)	0.012/13.09	15% (*n* = 3)	75% (*n* = 17)	0.014/6.84
	Severe	40.43% (*n* = 19)	59.57% (*n* = 28)		25.53% (*n* = 12)	74.47 (*n* = 35)		42.86% (*n* = 6) ^2^	57.14% (*n* = 8)		56.25% (*n* = 9)	43.75% (*n* = 7)	

^1^ Six PMM2-CDG patients with severe infections in the overall impact of these manifestations in QoL answered that they worsened other CDG signs/symptoms. ^2^ Two PMM2-CDG with severe allergies in the overall impact of these manifestations in QoL answered that they worsened other CDG signs/symptoms.

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
