# Peer review of "New Insights into Immunological Involvement in Congenital Disorders of Glycosylation (CDG) from a People-Centric Approach"

_jcm, 2020, doi:10.3390/jcm9072092_

Round 1

Reviewer 1 Report

The manuscript by Francisco et al is the first to address a totally neglected aspect in the CDG field: the immune system performance. In addition, they have chosen this patient centric angle that is really very intriguing and very much in line with the current thinking of modern medicine, in which the patient is in the lead and part of the discussions concerning his/her health.

So all in all I am really very fond of this manuscript. It is timely, clearly written and it will serve as pioneering piece of literature in the field.

I do have one comment, regarding the use of one publication as reference. This manuscript shows that allergies are significantly lower in PMM2-CDG participants. At row 848-849 the authors stated that glycans are important in allergen recognition and uptake and cite REF 71 (A. Al-Ghouleh, et al Glycosylation Pattern of Common Allergens : The Recognition and Uptake of Derp 1 by Epithelial and 1277 Dendritic Cells Is Carbohydrate Dependent, PLoS One. 7 (2012) e33929), saying that there it has been shown that glycan deficiency can decrease allergic responses. However, in that study is the allergen that lacks glycans and is less recognized, whereas in PMM2-CDG the allergen-sensing receptors would be lacking glycans. I think the authors should consider removing this statement or use a correct reference.

Author Response

Reply to the comments of the reviewer 1 on the manuscript entitled “New insights into immunological involvement in Congenital Disorders of Glycosylation (CDG) from a people-centric approach” with the Manuscript-ID jcm-833871

Reviewer 1

This manuscript shows that allergies are significantly lower in PMM2-CDG participants. At row 848-849 the authors stated that glycans are important in allergen recognition and uptake and cite REF 71 (A. Al-Ghouleh, et al Glycosylation Pattern of Common Allergens : The Recognition and Uptake of Derp 1 by Epithelial and 1277 Dendritic Cells Is Carbohydrate Dependent, PLoS One. 7 (2012) e33929), saying that there it has been shown that glycan deficiency can decrease allergic responses. However, in that study is the allergen that lacks glycans and is less recognized, whereas in PMM2-CDG the allergen-sensing receptors would be lacking glycans. I think the authors should consider removing this statement or use a correct reference.

Authors’ answer:

The authors have rewritten the text, framed REF 71 in a different way and added a new citation (REF 72). The text now reads: “Allergen glycan make-up has been shown to play an important role in recognition and uptake whereas the impact of host altered glycosylation has remained more elusive [71]. However, a recent study has shown that defective IgE glycosylation can decrease allergic responses [72]” line 897 to line 901.

Reviewer 2 Report

The study by Francisco R et al. using a people-centric approach, developed the questionnaire to better understand the role of the immune system in patients suffering from congenital disorders of glycosylation (CDG), and highlighted the role of gastrointestinal tract infections specifically in patients with PMM2-CDG. The specific comments are as followed.

  1. The manuscript is highly descriptive. Considering the clinical information derived from immune monitoring tests, they may need to discuss the implications of variations in the different immune subsets in patients with CDG on disease severity and its plausible and reported effects on different organs. For example, If there is any correlation between lymphocytosis, neutropenia, or monocytosis with GI tract infections or allergies in PMM2 or non-PMM2 CDG patients is unclear.
  2. Copy of questionnaire developed by the investigators for healthy controls (ImmunoHealthyQ) and patients (ImmunoCDGQ) may be helpful for the CDG research community and may be included in the supplemental material.
  3. On page #5, line 214, it is mentioned that 35 CDG patients were included. In the same paragraph and figure 2A, they mentioned that they included 209 patients with CDG, out of which 122 were PMM2-CDG and 87 were non-PMM2-CDG. It is unclear in which group of CDG (PMM2 vs non-PMM2) the 35 patients are included?
  4. The presentation of some of the figures including Fig.2B is poor, the size may be increased to provide a better picture of the frequencies of comorbidities present in patients with CDG. The numbers are overlapped in several figures, including Figures 2B, 7B, 7D, 8B.
  5. Figures 3 and 6 are missing.
  6. It is unclear why the number of patients excluded from the study (n=3 and n=5) is different at two different places? In the sentences, 364-365 on page 9, as mentioned “Participants who replied, “I don’t know” and “Dead” were excluded from this analysis (control n=1 and PMM2-CDG n=3”) and lines 369-370, as mentioned “while those who replied “I don’t know” and “Dead” were excluded from this analysis (PMM2-CDG n=5)”.
  7. Line 533, page 12: Authors have cited Figure 5B, however, there is no Fig. 5B in the manuscript.
  8. Figures 8A and 8B are repeated as part of the supplemental Figures S13A and S13B.

Minor Comments:

  1. Line 71, page #2: typographical error, “(…) a dynamic process through which the patients regulate the flow of information”.
  2. Please abbreviate the gastrointestinal tract as “GI” only where it is mentioned for the first time and use GI throughout the manuscript. Currently, its usage is inconsistent.

Author Response

Reply to the comments of the reviewer 2 on the manuscript entitled “New insights into immunological involvement in Congenital Disorders of Glycosylation (CDG) from a people-centric approach” with the Manuscript-ID jcm-833871

  1. The manuscript is highly descriptive. Considering the clinical information derived from immune monitoring tests, they may need to discuss the implications of variations in the different immune subsets in patients with CDG on disease severity and its plausible and reported effects on different organs. For example, If there is any correlation between lymphocytosis, neutropenia, or monocytosis with GI tract infections or allergies in PMM2 or non-PMM2 CDG patients is unclear.

Authors’ answer:

The authors agree that this is a topic that merits further research and elucidation, i.e., to be able to properly investigate the correlation between altered immune laboratorial cell counts (and immune cell/protein dysfunction) and the presence/absence of clinical immune manifestations - such as infections or allergies - or even between overall CDG severity. It is indeed paramount to establish if/how these cellular alterations contribute to PMM2-CDG clinical phenotypes. However, in our study only a small subset of participants identified any permanent laboratorial immune alterations which hinders the possibility to perform such analysis (e.g. as you can see on Table S15 the maximum number of PMM2-CDG patients reporting any constant alteration are 5 patients who reported constant leukocytosis) and draw valid conclusions when such small numbers are present.

 In summary, the authors agree with the extreme relevance of this poorly known topic and find that its elucidation can have an important impact in the understanding of PMM2-CDG physiopathology (and probably in other CDG as well). We addressed this also in the discussion (Line 963 to line 966) as we felt our study could not properly answer these questions but acknowledge their relevance.

  1. Copy of questionnaire developed by the investigators for healthy controls (ImmunoHealthyQ) and patients (ImmunoCDGQ) may be helpful for the CDG research community and may be included in the supplemental material.

Authors’ answer:

The authors agree that making the questionnaires available can be useful for other researchers dedicated to CDG. As the original format of these surveys is electronic (Survey Monkey) we believe it is most appropriate to share them in their original format, mostly because of the logic that has been applied to various questions and which is mandatorily lacking in the paper format. Also, since the questionnaires are available in several different languages making them available in their e-version allows for easy and full sharing of very language version. Hence, we have added to the manuscript: lines 143 – 146 the following “An electronic copy of the ImmunoCDGQ (in all language versions) is available at https://www.researchcdg.com/immunocdgq.html, while the ImmunoHealthyQ (also all languages) is publicly available at  https://www.researchcdg.com/immunoq.html. “.

  1. On page #5, line 214, it is mentioned that 35 CDG patients were included. In the same paragraph and figure 2A, they mentioned that they included 209 patients with CDG, out of which 122 were PMM2-CDG and 87 were non-PMM2-CDG. It is unclear in which group of CDG (PMM2 vs non-PMM2) the 35 patients are included?

Authors’ answer:

The authors are referring to the 35 CDG types, not patients, as it is represented in Figure 1A. However, we understand why this was not clearly understood. Hence, for better comprehension the localization of such statement has been altered the paragraph that now reads “Our questionnaires assessed 209 CDG patients and 349 healthy participants. PMM2-CDG was the most represented CDG (58.4 %, n=122/209) and included 4 deceased patients (Supplementary table 4). In total 35 different CDG were included. After PMM2-CDG, ALG6-CDG (MIM:603147) (4.3 %, n=9/209) and SLC35A2-CDG (MIM: 300896) (3.3 %, n=7/209) were the most reported (Figure 1A). This prevalence echoes published data on these CDG [6], [12] [36].  All these CDG were clustered and herein referred to as the non-PMM2-CDG group (Figure 1A)” from lines 218 to 221.

  1. The presentation of some of the figures including Fig.2B is poor, the size may be increased to provide a better picture of the frequencies of comorbidities present in patients with CDG. The numbers are overlapped in several figures, including Figures 2B, 7B, 7D, 8B.

Authors’ answer:

The authors have improved the size and lettering of the Figures (particularly of that of Fig. 2B) according to the reviewers suggestions.

  1. Figures 3 and 6 are missing.

Authors’ answer:

The authors apologize for this technical error and have now added these figures to the manuscript body. Additionally, all figures are available as individual files in JPEG format and meeting the resolution limitations defined by JCM.

  1. It is unclear why the number of patients excluded from the study (n=3 and n=5) is different at two different places? In the sentences, 364-365 on page 9, as mentioned “Participants who replied, “I don’t know” and “Dead” were excluded from this analysis (control n=1 and PMM2-CDG n=3”) and lines 369-370, as mentioned “while those who replied “I don’t know” and “Dead” were excluded from this analysis (PMM2-CDG n=5)”.

Authors’ answer:

These differences  in the number of participants that were excluded from the analysis in each questions are directly related to the differences in the number participants who answered “I don’t know” in each of the questions, as the questions were independent from each other. Upon re-reading the text, the authors felt that adding the numbers of patients excluded from the analysis, besides the already stated reasons for exclusion, hindered the reading and understanding of the message.For these reasons we opted to omit this information keeping only the exclusion criteria. We have uniformly adopted this criterion throughout the entire manuscript.

  1. Line 533, page 12: Authors have cited Figure 5B, however, there is no Fig. 5B in the manuscript.

Authors’ answer:

The authors apologize for this mistake. The correct figure (Fig. 3I) has been indicated in the line 556.

  1. Figures 8A and 8B are repeated as part of the supplemental Figures S13A and S13B.

Authors’ answer:

The differences between Figures 8A and 8B and Figures S13A and S13B are that in the supplementary figures the non-PMM2-CDG group is also represented in addition. We opted to maintain in the main manuscript figures illustrating PMM2-CDG findings as far as the most specific clinical results are concerned. However, we felt that we should also show the findings related to the non-PMM2-CDG group in the supplementary material that is why we opted to include these figures.

Minor Comments:

  1. Line 71, page #2: typographical error, “(…) a dynamic process through which the patients regulate the flow of information”.

Authors’ answer:

Upon revision, the authors decided to replace the existing ipis verbis definition of Patient-centricity as defined in REF 20 and adapted the concepts explained in REF 20 in their own words. The text from line 71 to line 75 now reads “Patient or people centricity could be explained as a dynamic process which involves the patients/citizens as equal partners by continuously valuing their insights, preferences, values, and beliefs”

  1. Please abbreviate the gastrointestinal tract as “GI” only where it is mentioned for the first time and use GI throughout the manuscript. Currently, its usage is inconsistent.

Authors’ answer:

As suggested by the reviewer gastrointestinal tract as been presented as GI consistently throughout the text with the exception of its first reference in the abstract and introduction and as a keyword.